# *Clonorchis sinensis* on the prognosis of patients with spontaneous rupture of Hepatocellular Carcinoma: An inverse probability of treatment weighting analysis

**Hang-Hang Ni**[1,2☯], **Zhan Lu**[1☯], **Cheng-Lei Yang**[1☯], **Yu-Ting Lv**[1], **Chun-Xiu Lu**[1], **Bang-De Xiang**[1,3]*

1 Department of Hepatobiliary Surgery, Guangxi Medical University Cancer Hospital, Nanning, People's Republic of China, 2 Department of Hepatobiliary Surgery, Zhongshan City People's Hospital, Zhongshan, People's Republic of China, 3 Key Laboratory of Early Prevention and Treatment for Regional High-Frequency Tumors, Ministry of Education, Nanning, People's Republic of China

☯ These authors contributed equally to this work.
* xiangbangde@gxmu.edu.cn

## Abstract

### Background

We examined the impact of the *Clonorchis sinensis* (*C. sinensis*) infection on the survival outcomes of spontaneous rupture Hepatocellular Carcinoma (srHCC) patients undergoing hepatectomy.

### Methods

Between May 2013 and December 2021, 157 consecutive srHCC patients who underwent hepatectomy were divided into an no *C. sinensis* group (n = 126) and *C. sinensis* group (n = 31). To adjust for differences in preoperative characteristics an inverse probability of treatment weighting (IPTW) analysis was done, using propensity scores. Overall survival (OS) and recurrence-free survival (RFS) were compared before and after IPTW. Multivariate Cox regression analysis was performed to determine whether the *C. sinensis* infection was an independent prognostic factor after IPTW.

### Results

In original cohort, the no *C. sinensis* group did not show a survival advantage over the *C. sinensis* group. After IPTW adjustment, the median OS for the *C. sinensis* group was 9 months, compared to 29 months for the no *C. sinensis* group. *C. sinensis* group have worse OS than no *C. sinensis* group (p = 0.024), while it did not differ in RFS(p = 0.065). The multivariate Cox regression analysis showed that *C. sinensis* infection and lower age were associated with worse OS.

**Funding:** BDX received the financial support from the National Natural Science Foundation of China (grant no. 81960450), the National Major Special Science and Technology Project (grant no. 2017ZX10203207). The funders had no role in study design, data collection and analysis, decision to publish, or preparation of the manuscript.

**Competing interests:** The authors have declared that no competing interests exist.

## Conclusions

The *C. sinensis* infection has an adverse impact on os in srHCC patients who underwent hepatectomy.

## Author summary

*Clonorchis sinensis*, a trematode of the Opisthorchiidae family, is a genus of zoonotic parasite and liver fluke. This species specifically resides in the bile ducts of humans and mammals during its adult stage. Known as a class I carcinogen by the International Agency for Research on Cancer, *Clonorchis sinensis* has gained recognition for its association with cholangiocarcinoma. The occurrence of *Clonorchis sinensis* and its transmission to humans heavily relies on geographical factors and the dietary preferences of its hosts. Individuals, along with other definitive hosts, contract the parasite by consuming raw or undercooked freshwater fish or shrimp that harbor infective metacercariae. A study indicated that patients with hepatocellular carcinoma combined with *Clonorchis sinensis* infection have a worse prognosis after hepatectomy. This study aimed to determine the prognostic significance of *Clonorchis sinensis* infection with spontaneous rupture Hepatocellular Carcinoma(srHCC). We have retrospectively analyzed the infection rate of srHCC. We found that 19.7% of patients with srHCC had a combination of *Clonorchis sinensis* infection. We also found that srHCC patients with *Clonorchis sinensis* infection have a worse prognosis after hepatectomy. This is a single center study, hopefully lending guidance the prognosis of srHCC with *Clonorchis sinensis*.

## Introduction

Hepatocellular carcinoma (HCC) is the sixth most common cancer and the third leading cause of cancer-related deaths worldwide and with the highest incidence in Asia, especially in China [1]. Spontaneous rupture of hepatocellular carcinoma (srHCC) is a special type of HCC with the incidence of 2.3–26% [2,3], and mortality is 25–75% [4]. The mechanisms of srHCC remain unclear. Possible reasons include large tumor size, ischemic necrosis, and vascular compression caused by rapid tumor growth [5–7].

Clonorchiasis, which is caused by the parasitic zoonosis known as *Clonorchis sinensis* (*C. sinensis*), is a major health concern in Asia. It is estimated that around 35 million individuals are infected with this disease, with approximately 15 million of these cases being reported in China alone [8–10]. The region most severely impacted by this affliction are Guangdong Province and the Guangxi Zhuang Autonomous Region in Southern China [11–13]. *C. sinensis* transmission occurs when individuals ingest raw or undercooked fish that is contaminated with *C. sinensis* metacercariae [14]. One study confirmed that *C. sinensis* could aggravate the progression of liver fibrosis [15]. Our study demonstrated that *C. sinensis* infection increases risk of HCC in a rat model [16]. Our clinical study also revealed that HCC with *C. sinensis* infection experience a poor prognosis after hepatectomy [17]. However, no study has reported an association between *C. sinensis* infection and the prognosis of srHCC. To our knowledge, the prognosis of srHCC remain controversial as yet, only a small number of studies have been conducted to determine the prognostic factors for postoperative survival. Therefore, it is necessary to study whether *C. sinensis* infection affects the prognosis of srHCC after hepatectomy.

Observational studies, which gather data from routine clinical practice, offer valuable insights into the effects of treatments on a broader range of patients and over a longer period of time. However, these studies may be susceptible to confounding by indications, as inherent differences between treatment groups can make it difficult to reliably estimate the treatment effect by directly comparing outcomes [18]. To address this issue, the use of inverse probability of treatment weighting (IPTW) has gained popularity. This statistical approach, based on propensity scores, has been demonstrated to result in more robust and less biased estimates of treatment effects [19].

In this study, we therefore conducted this retrospective study to analyze *C. sinensis* associated with the recurrence free survival (RFS) and overall survival (OS) of HCC patients with spontaneous rupture after hepatectomy via IPTW method.

## Materials and methods

### Ethics statement

The study was conducted according to the principles of the Declaration of Helsinki and was approved by the Ethics Committee of the Affiliated Cancer Hospital of Guangxi Medical University (Approval No.: LW2024004). Individual consent for this retrospective analysis was waived.

### Study population

This is a retrospective study at the Affiliated Cancer Hospital of Guangxi Medical University in Nanning, Guangxi Zhuang Autonomous Region, China, between May 2013 and December 2021. Of 328 patients with ruptured HCC, 171 were excluded (17 with previous treatment, 95 receiving TAE or TACE, 33 without treatment, and 26 without complete follow up data). A total of 157 patients were included in the analysis, of which 31 patients with *C. sinensis* and 126 patients without *C. sinensis*. (**Fig 1**).

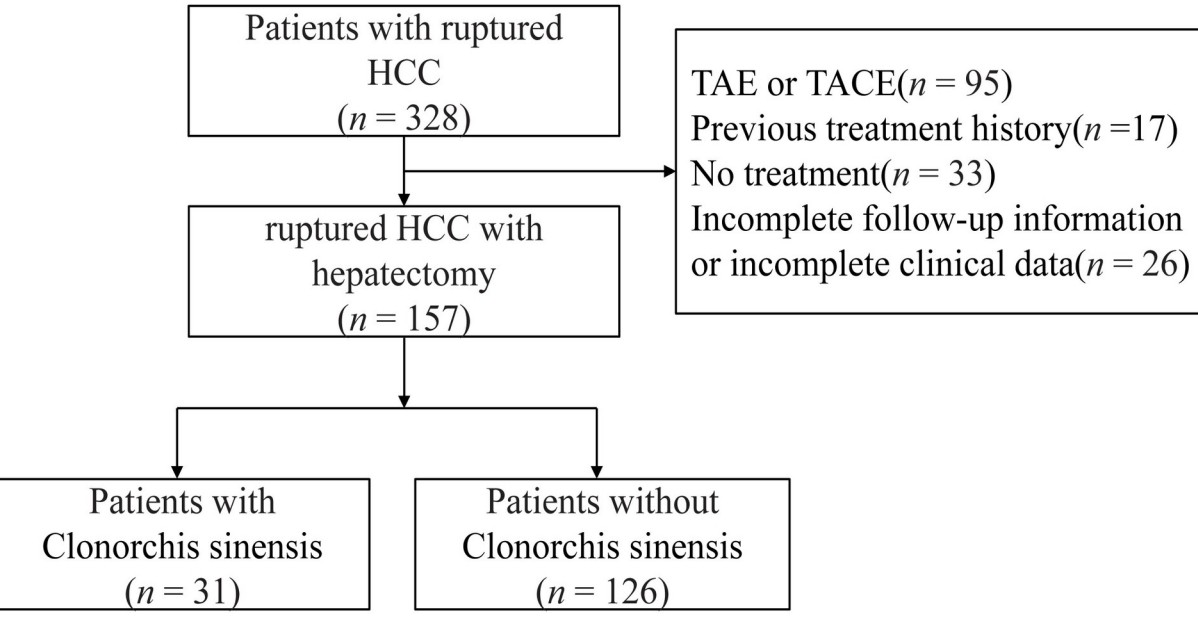

**Fig 1. Study flowchart.** TACE, transcatheter arterial chemoembolization; TAE, transcatheter arterial embolization.

## Definition

The diagnostic criteria for clonorchiasis are specified below, and the confirmation of *Clonorchis Sinensis* diagnosis required the presence of any one factor [14,20,21]. (1) Clinical diagnosis cases: In order to qualify for this type of diagnosis, the patient must have a history of eating raw or half-raw freshwater fish or shrimp, and should have a record of living, working, or traveling in endemic areas. Additionally, there should be a noticeable but mild expansion in the intrahepatic bile ducts, which can be detected through imaging (CT, MRI, or ultrasonography can determine the pattern of bile duct expansion). It should be noted that any locally induced expansion due to factors such as bile duct stones, cancerous embolism, tumor compression, or thrombosis will not be considered. (2) The presence of adult *C. sinensis* in the liver or gallbladder is confirmed through intraoperative or postoperative pathological examination. (3) Preoperative fecal examination has given a positive result in detecting *C. sinensis* eggs. The diagnosis of srHCC relies on the manifestation of symptoms that include acute abdominal pain and signs of peritonitis, which will be confirmed through contrast-enhanced CT or MRI scans that detect the extravasation of contrast materials from the tumor's surrounding perihepatic hematoma and signs of peritoneal effusion. When necessary, diagnostic abdominal paracentesis is performed to further validate the diagnosis. The presence of srHCC will be confirmed by intraoperative observations of tumor rupture with hemoperitoneum and postoperative pathology, or contrast medium extravasation in angiography.

## Treatment

The objective of performing partial hepatectomy was to remove all detectable cancerous tissue. All operative procedures were performed by experienced surgeons, and the abdominal cavity was extensively explored during the operation. The desirable tumor margin during resection was more than 1 cm. To prevent the spread of tumor cells, a significant quantity of distilled water was used to cleanse the peritoneum. Anthelmintic therapy was not administered to srHCC patients with *C. sinensis* before or after surgery, and deworming was only done if the patient volunteered for anthelmintic therapy.

## Patient Follow-Up

Patients were followed-up at one month after the operation and subsequently every three months for the first two years, every six months thereafter. Laboratory examinations (serum AFP, liver function, blood tests), abdominal ultrasonography, and contrast-enhanced CT were collected. Follow-up period was terminated on March, 2023. The diagnosis of recurrence was based on typical findings of HCC on CT or MR imaging.

## Statistical analysis

To minimize the potential bias caused by confounding factors, we utilized stabilized inverse probability of treatment weights (IPTW), a propensity score-based method used to balance baseline variables without sample loss. To achieve this, we employed a logistic regression model that encompassed all available covariates. We aimed to prevent an unintended imbalance in other parameters that may not directly correlate with the probability of receiving the treatment but could still have an unknown impact on the outcome. The obtained propensity score was then used to generate IPTW through appropriate math, which were used to weight each clinical feature, as well as measured outcomes, of each patient in both treatment groups [22,23].

RFS was measured from the date of hepatectomy until tumor recurrence. Overall survival (OS) was measured between the date of hepatectomy and the date of death or the date of the last follow-up visit. RFS and OS was evaluated using the Kaplan–Meier method and compared with the log-rank test. The variables with statistical significance were firstly identified by the univariate Cox regression ($P<0.05$), and then were analyzed by the multivariate cox regression. Statistical analyses were performed using SPSS 26.0 (IBM, Chicago, IL, USA) and R software version 4.1.2 (http://www.R-project.org), and a two-sided $p< 0.05$ denoted statistically significant.

## Results

### Baseline characteristics

A total of 157 patients were included in this study; of which 31(19.7%) patients with *C. sinensis*, 126(80.3%) without *C. sinensis*. There were 135(80.0%) patients with HbsAg and 22(20.0%) without. The median age is 50 years. This study included 133 (84.7%) men and 24 (15.3%) women. **Table 1** shows the baseline characteristics of the two study groups before and after IPTW. In the original cohort, there were significant differences with respect to male, HbsAg, Liver cirrhosis, Edmondson stage, AST, PLT, Tumor number and MVI(SMD>0.2). A good covariate balance was achieved after the application of IPTW; the only exception was HbsAg (SMD >0.2).

### Overall survival and recurrence-free survival before matching

**Fig 2** shows the OS and RFS curves for the patients with or without *C. sinensis* before matching. The median OS for the *C. sinensis* group was 17 months while that for the no *C. sinensis* group was 32 months. The 1, 3 and 5-year OS rates for the *C. sinensis* group were 64.5%, 39.6%

**Table 1. Baseline characteristics of the study population before and after IPTW adjustment.**

| Variable | Original Cohort | | | After IPTW | | |
|---|---|---|---|---|---|---|
| | No *C. sinensis* (n = 126) | *C. sinensis* (n = 31) | SMD | No *C. sinensis* (n = 155.8) | *C. sinensis* (n = 154.0) | SMD |
| Male | 103 (81.7) | 30 (96.8) | 0.500 | 131.7 (84.6) | 133.6 (86.7) | 0.062 |
| Age (≤50 years) | 68 (54.0) | 14 (45.2) | 0.177 | 80.0 (51.4) | 80.4 (52.2) | 0.017 |
| HbsAg(positive) | 111 (88.1) | 24 (77.4) | 0.285 | 135.1 (86.7) | 120.0 (77.9) | 0.233 |
| Liver cirrhosis(positive) | 86 (68.3) | 15 (48.4) | 0.411 | 101.5 (65.1) | 99.6 (64.7) | 0.010 |
| PT(≥13s) | 55 (43.7) | 11 (35.5) | 0.168 | 66.2 (42.5) | 80.0 (52.0) | 0.191 |
| ALB(≥35 g/L) | 80 (63.5) | 22 (71.0) | 0.160 | 99.4 (63.8) | 91.9 (59.7) | 0.085 |
| AFP (≥200 ng/mL) | 79 (62.7) | 22 (71.0) | 0.176 | 100.6 (64.6) | 101.8 (66.1) | 0.033 |
| TBIL(≥17.1μmol/L) | 65 (51.6) | 18 (58.1) | 0.130 | 80.0 (51.4) | 74.4 (48.3) | 0.061 |
| Edmondson III+IV | 91 (72.2) | 26 (83.9) | 0.284 | 115.6 (74.2) | 114.4 (74.3) | 0.001 |
| AST(≥40U/L) | 90 (71.4) | 17 (54.8) | 0.349 | 106.7 (68.5) | 112.5 (73.0) | 0.099 |
| ALT(≥40 U/L) | 54 (42.9) | 11 (35.5) | 0.151 | 64.0 (41.1) | 55.1 (35.8) | 0.109 |
| PLT(≥100×10⁹/L) | 123 (97.6) | 31 (100.0) | 0.221 | 152.8 (98.1) | 154.0 (100.0) | 0.198 |
| Child Pugh B | 34 (27.0) | 7 (22.6) | 0.102 | 41.5 (26.7) | 53.3 (34.6) | 0.173 |
| Tumor size (≤10 cm) | 77 (61.1) | 17 (54.8) | 0.127 | 93.4 (59.9) | 85.0 (55.2) | 0.096 |
| Tumor number (>1) | 39 (31.0) | 6 (19.4) | 0.270 | 44.2 (28.4) | 38.0 (24.7) | 0.084 |
| MVI(positive) | 74 (58.7) | 23 (74.2) | 0.332 | 95.9 (61.6) | 107.1 (69.6) | 0.169 |

SMD is the difference in mean or proportion divided by the pooled standard error; imbalance is defined as an absolute value greater than 0.2.

**Abbreviations:** AFP, alpha-fetoprotein; ALT, alanine aminotransferase; ALB, albumin; AST, aspartate aminotransferase; HbsAg, hepatitis B surface antigen; MVI, microvascular invasion; PLT, platelet; PT, prothrombin time; TBIL, total bilirubin.

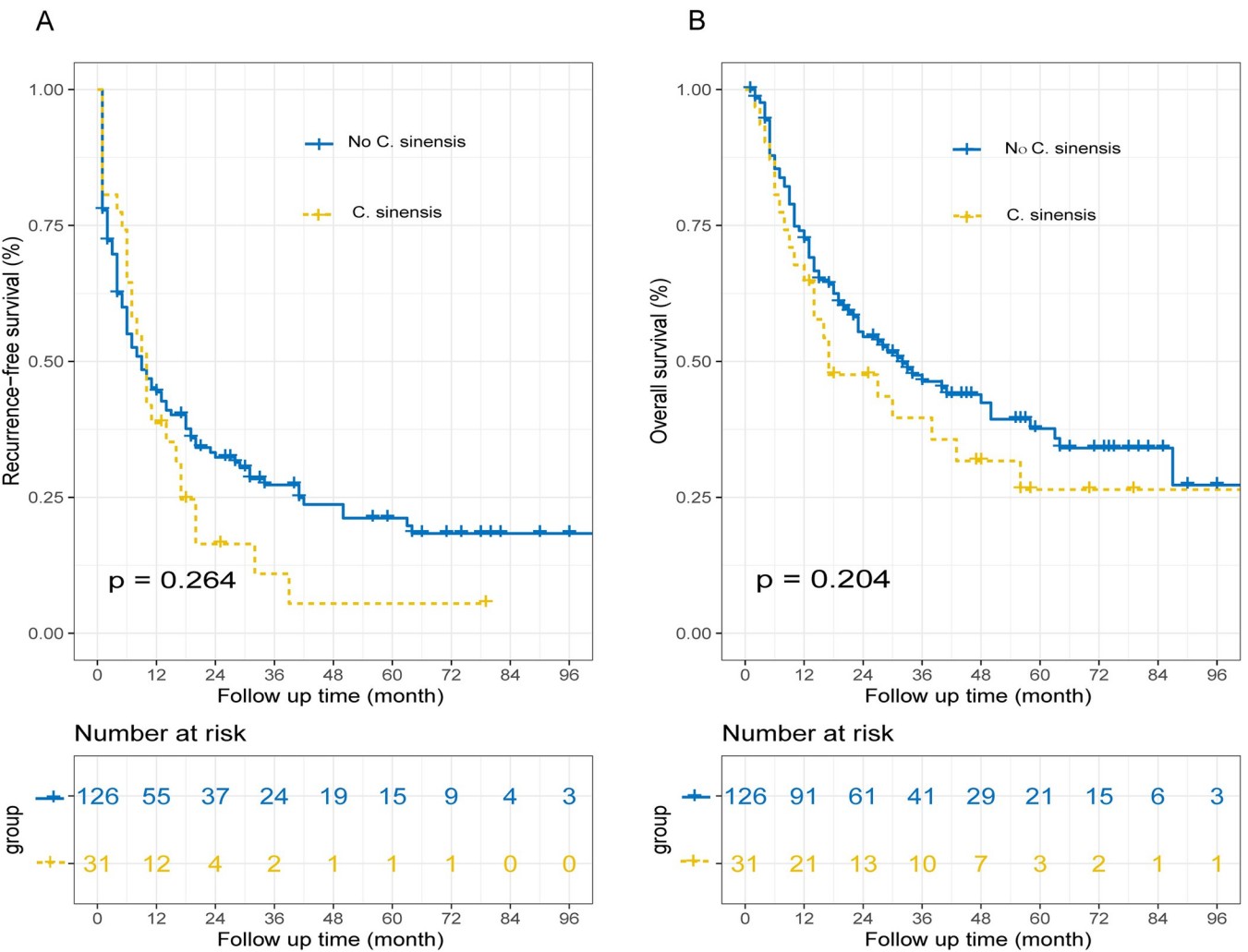

**Fig 2.** Recurrence-free survival (RFS) and Overall survival (OS) curves before matching. The RFS curve (A) and the OS curve (B).

and 26.4%, respectively, whereas those for the no *C. sinensis* group were 72.4%, 46.3% and 37.6%, respectively, with no significant difference between the two groups ($p = 0.204$). In terms of RFS, the median for the *C. sinensis* group was 10 months, while that for the no *C. sinensis* group was 9 months. The 1, 3 and 5-year RFS rates for the *C. sinensis* group were 38.7%, 10.9%, and 5.5% respectively, while those for the no *C. sinensis* group were 44.4%, 27.3% and 21.2%, respectively. However, there was no significant difference between the two groups ($p = 0.264$).

## Overall survival and recurrence-free survival after inverse probability of treatment weighting analysis

After adjusting for potential confounders using inverse probability of treatment weighting (IPTW), following the analysis, the median RFS for the *C. sinensis* group was noted to be 6 months, while that for the no *C. sinensis* group was observed to be 8 months, there was no significant difference in RFS between the two groups ($p = 0.065$). Similarly, the median OS for the *C. sinensis* group was found to be 9 months, compared to 29 months for the no *C. sinensis* group, there was significant difference in OS between the two groups ($p = 0.024$) (**Fig 3**).

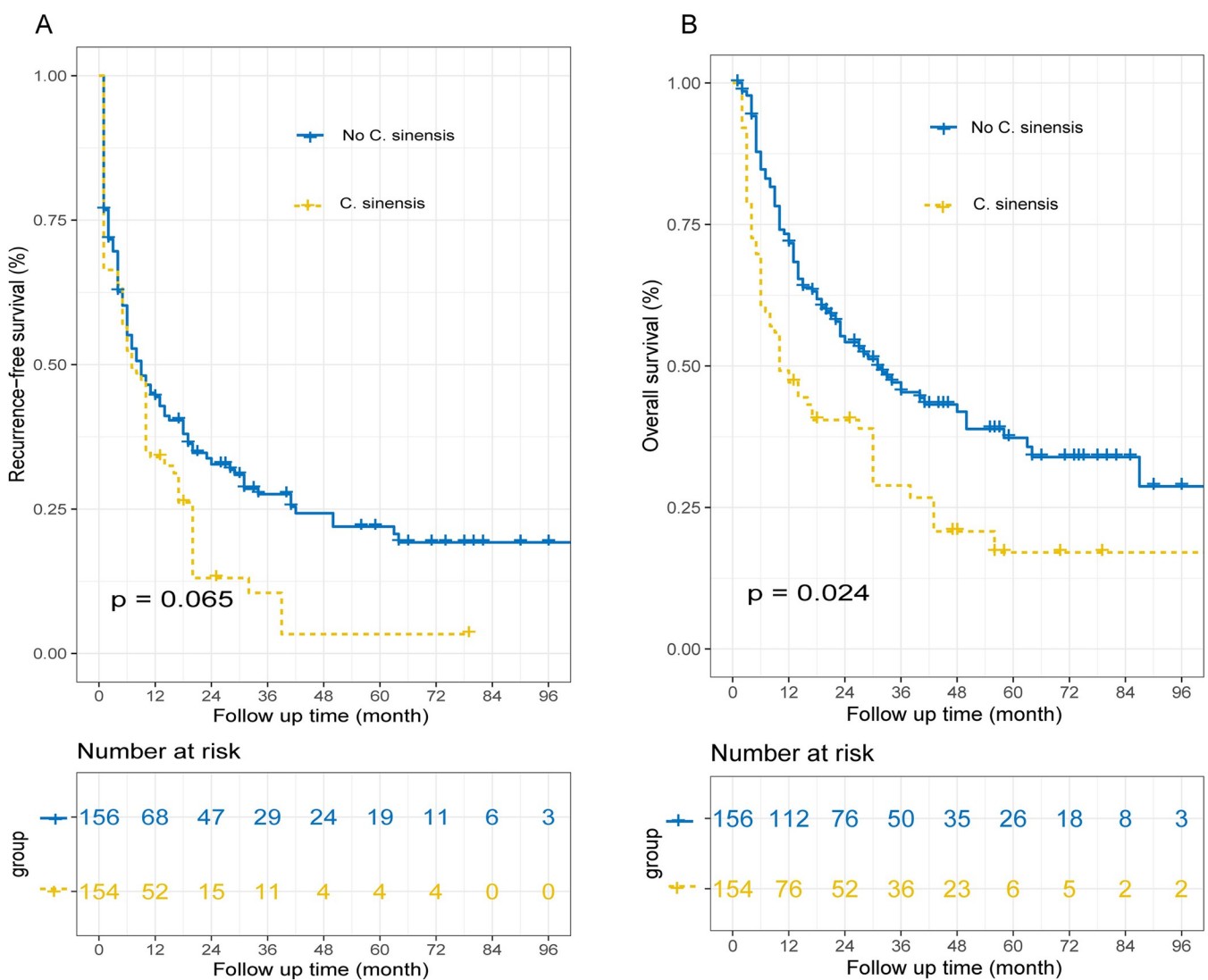

**Fig 3. Recurrence-free survival (RFS) and Overall survival (OS) curves after inverse probability of treatment weighting analysis.** The RFS curve (A) and OS curve (B).

### *C. sinensis* and HBV co-infection Prognostic analysis

24 patients were co-infected with *C. sinensis* and HBV, the other 133 patients included 7 *C. sinensis* infections alone, 111 HBV infections alone, and 15 double negative patients. There was no significant difference in RFS between the *C. sinensis*-positive + HBV-positive group and others group before and after IPTW. Similarly, there was no significant difference in OS between the two groups before and after IPTW (**Fig 4**).

### Prognostic factors associated with Overall survival after inverse probability of treatment weighting

After IPTW weighting, *C. sinensis* infection and Age ≤50 years were shown to be associated with Overall survival risk in the univariate analysis($p<0.05$). Multivariate Cox regression analyses demonstrated that *C. sinensis* infection (HR = 1.930) and Age ≤50 years (HR = 1.869) were independent prognostic factors for OS($p<0.05$). (Table 2)

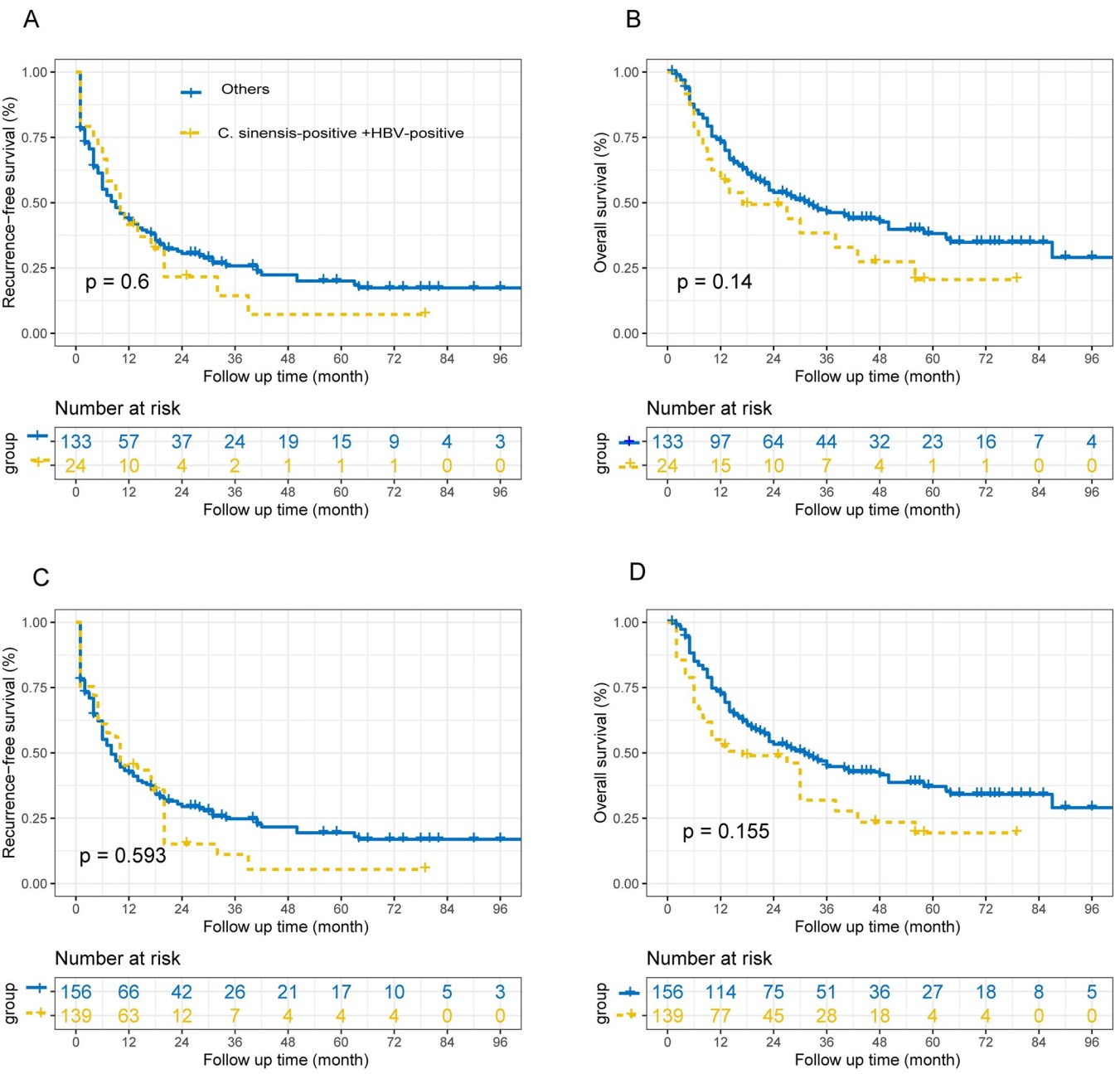

**Fig 4. Overall survival (OS) and recurrence-free survival (RFS) curves before and after inverse probability of treatment weighting analysis.** The PFS curve (A) and OS curve(B) before inverse probability of treatment weighting analysis. The PFS curve(C) and OS curve (D) after inverse probability of treatment weighting analysis. *C. sinensis*, Clonorchis sinensis; HBV, Hepatitis B virus.

## Discussion

Supported by IPTW and multivariate Cox regression analyses, our findings suggest that *C. sinensis* infection serves as an independent prognostic indicator for OS in patients with srHCC, and is associated with increased mortality rates. While no significant impact on RFS was observed, a clear trend towards higher relapse rates was noted. The weight assigned to each patient in the cohort is determined by their probability of exposure to the *C. sinensis*

**Table 2. Univariate and multivariable associations between risk factors and overall survival in patients after IPTW analysis.**

| Variable | Univariate analysis | | | Multivariate analysis | | |
|---|---|---|---|---|---|---|
| | HR | 95%CI | *P* | HR | 95%CI | *P* |
| gender (M/F) | 0.603 | 0.214–1.694 | 0.337 | | | |
| Age (≤50 years) | 1.771 | 1.046–3.001 | 0.033 | 1.869 | 1.053–3.316 | 0.032 |
| Liver cirrhosis (positive) | 0.952 | 0.502–1.807 | 0.880 | | | |
| PT(≥13s) | 1.478 | 0.840–2.600 | 0.176 | | | |
| HbsAg(positive) | 0.760 | 0.267–2.161 | 0.607 | | | |
| ALB(≥35g/L) | 0.879 | 0.435–1.774 | 0.718 | | | |
| AFP (≥200 ng/mL) | 1.331 | 0.752–2.356 | 0.327 | | | |
| TBIL(≥17.1μmol/L) | 0.782 | 0.440–1.391 | 0.403 | | | |
| Edmondson III+IV | 1.583 | 0.805–3.114 | 0.183 | | | |
| AST(≥40 U/L) | 1.463 | 0.895–2.392 | 0.130 | | | |
| ALT(≥40 U/L) | 1.168 | 0.633–2.155 | 0.619 | | | |
| PLT(≥100×10$^9$/L) | 0.996 | 0.415–2.393 | 0.993 | | | |
| Child Pugh B | 1.125 | 0.476–2.658 | 0.788 | | | |
| Tumor size (≤10 cm) | 0.760 | 0.406–1.422 | 0.390 | | | |
| Tumor number (≥1) | 1.167 | 0.597–2.279 | 0.652 | | | |
| MVI (positive) | 1.330 | 0.806–2.194 | 0.264 | | | |
| *Clonorchis sinensis*(positive) | 1.836 | 1.083–3.113 | 0.024 | 1.930 | 1.101–3.384 | 0.022 |

**Abbreviations:** AFP, alpha-fetoprotein; ALT, alanine aminotransferase; ALB, albumin; AST, aspartate aminotransferase; HbsAg, hepatitis B surface antigen; MVI, microvascular invasion; PLT, platelet; PT, prothrombin time; TBIL, total bilirubin.

being studied. Once the treatment groups are balanced, applying weights during statistical tests or regression models helps mitigate or eliminate the influence of known confounding factors.

To the best of our knowledge, no previous cohort study has investigated the association between *C. sinensis* infection and srHCC in the overworld population. *C. sinensis* are prevalent in our region. One study reported that *C. sinensis* is an important risk factor for ICC and HCC in a high prevalence area [24]. Prior research indicated that certain molecules have the potential to significantly inhibit apoptosis, as well as stimulate the proliferation and migration of human HCC cells. These effects may potentially exacerbate the progression of HCC in patients who are also infected with *C. sinensis* [25,26]. Patients diagnosed with HCC and co-existing *C. sinensis* infection typically face a grim outlook following hepatectomy. This study revealed that *C. sinensis* infection does not affect RFS and OS with surgery for srHCC before IPWT. However, after IPTW, patients with srHCC infected with *C. sinensis* have a worse OS. The most likely reason is that before IPTW matching, the baseline characteristics of the patients between two group was unbalanced. Our pervious study has demonstrated that patients with HCC and *C. sinensis* infection experience a poor prognosis after hepatectomy, but *C. sinensis* is not an independent risk factor affecting OS [17]. This study demonstrated that *C. sinensis* infection is an independent prognostic factor affecting the OS of patients with srHCC. The utilization of the IPTW technique, which is a comparatively innovative approach in the area under discussion, produced a weighted sample that has a more balanced distribution of covariates between the *C. sinensis* group and the no *C. sinensis* group. This allowed us to not only minimize the discrepancies in patient features but also maintain the sample size and statistical potency, leading to a more refined and accurate assessment of the treatment impact.

In China, individuals with HBV infections are frequently located in regions with a high occurrence of *C. sinensis* [24,27]. The hepatitis B virus is serious pathogen that can cause liver

disease and cancer. Study demonstrated that the simultaneous presence of HBV and *C. sinensis* could have a detrimental effect on liver function and potentially facilitate the proliferation of HBV [28]. This study showed that co-infection of *C. sinensis* and HBV did not affect the prognosis of patients, possibly because most of the patients in the study were infected with HBV. *C. sinensis* infection may only promote HBV proliferation, and *C. sinensis* cannot directly affect the prognosis of hepatocellular carcinoma.

Age was found to be a significant parameter associated with survival in our study. Specifically, we found that srHCC patients with lower age had a worse prognosis. This finding is consistent with previous research results [29–31]. Some studies have found that young HCC patients have better prognosis because they have better liver function [32–35]. However, others have reported that young patients with advanced tumor factors, have a poor prognosis [36–38]. According to the current eighth edition of the AJCC/UICC (American Joint Committee on Cancer/Union for International Cancer Control) classification [39], srHCC is defined as a T4 stage tumor, which is an advanced tumor factor. Our research results also suggest that age is an independent risk factor affecting OS of srHCC patients.

This study has some limitations. Firstly, due its retrospective nature, and the confounding effects between the two groups could not be completely excluded after IPTW analyses. On the other hand, because the underlying liver disease in the present study was chronic hepatitis B infection in most patients, the results should be validated in other study groups with HCV infections or alcohol consumption. Thirdly, this is a single center and small sample analysis, and selection bias was present due to the lack of external validation from another constitution.

In conclusion, our study demonstrated that *C. sinensis* infection and low age (<50 years) were independent predictors for OS of srHCC patients after hepatectomy. Our results provide important clinical information for srHCC with *C. sinensis* infection.

## Author Contributions

**Conceptualization:** Hang-Hang Ni.

**Data curation:** Hang-Hang Ni, Yu-Ting Lv.

**Formal analysis:** Cheng-Lei Yang, Chun-Xiu Lu.

**Funding acquisition:** Bang-De Xiang.

**Investigation:** Hang-Hang Ni, Zhan Lu.

**Methodology:** Cheng-Lei Yang.

**Project administration:** Bang-De Xiang.

**Software:** Hang-Hang Ni, Zhan Lu.

**Validation:** Cheng-Lei Yang.

**Visualization:** Hang-Hang Ni.

**Writing – original draft:** Hang-Hang Ni, Zhan Lu.

**Writing – review & editing:** Bang-De Xiang.

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
