## [Decision Letter · Decision Letter 0]

20 Oct 2023

Dear Dr. Xiang,

Thank you very much for submitting your manuscript "Clonorchis sinensis on the prognosis of patients with spontaneous rupture of Hepatocellular Carcinoma: an inverse probability of treatment weighting analysis" for consideration at PLOS Neglected Tropical Diseases. As with all papers reviewed by the journal, your manuscript was reviewed by members of the editorial board and by several independent reviewers. In light of the reviews (below this email), we would like to invite the resubmission of a significantly-revised version that takes into account the reviewers' comments. 

We cannot make any decision about publication until we have seen the revised manuscript and your response to the reviewers' comments. Your revised manuscript is also likely to be sent to reviewers for further evaluation.

Sincerely,

Javier Sotillo

Academic Editor

Uriel Koziol

Section Editor

Reviewer's Responses to Questions

**Key Review Criteria Required for Acceptance?**

**Methods**

-Are the objectives of the study clearly articulated with a clear testable hypothesis stated?

-Is the study design appropriate to address the stated objectives?

-Is the population clearly described and appropriate for the hypothesis being tested?

-Is the sample size sufficient to ensure adequate power to address the hypothesis being tested?

-Were correct statistical analysis used to support conclusions?

-Are there concerns about ethical or regulatory requirements being met?

Reviewer #1: The paragraph "Study Population" should be re-written like this: Of 328 patients with ruptured HCC, 171 were excluded (17 with previous treatment, 95 receiving TAE or TACE, 33 without treatment, and 26 without complete follow up data). In addition, the prognosis-related clinical factors including MVI, BCLC stage, tumor size, and HBV DNA should be compared between 157 included patients and 171 excluded patients. This will let readers know if the 157 patients represent the 328 patients in total. Others are OK.

Reviewer #2: (No Response)

Reviewer #3: 1. It would be better if the authors can provide the information of IPTW in the introduction or method, including the advantages, and how it important and improve the result or previous reports. 

2. To obvious, please describe separately between the baseline characteristics and the result that comparison of before and after using IPTW. Importantly, please use Chi-square to compare the different between before and after using IPTW. Could you please explain more about the statistical analysis of this part?

**Results**

-Does the analysis presented match the analysis plan?

-Are the results clearly and completely presented?

-Are the figures (Tables, Images) of sufficient quality for clarity?

Reviewer #1: The results are fully and clearly presented. Tables and Figs need to be polished. HBV infection between the two groups should be included in the survival analysis because HBV replication has been proven to be an independent risk factor of HCC prognosis.

Reviewer #2: (No Response)

Reviewer #3: 1. Are the results clearly and completely presented in Table 1?

2. This finding did not show the significant data of C. sinensis and no C. sinensis group when no using IPTW, could you please explain and discussion in this point.

**Conclusions**

-Are the conclusions supported by the data presented?

-Are the limitations of analysis clearly described?

-Do the authors discuss how these data can be helpful to advance our understanding of the topic under study?

-Is public health relevance addressed?

Reviewer #1: The significance was found in patients adjusted by inverse probability of treatment weighting (IPTW). Pls discuss how this kind of matching might avoid potential bias?

Reviewer #2: (No Response)

Reviewer #3: 1. Please elucidate how srHCC differ from HCC, what is the significant point to study? 

2. Please add more risk factors of HCC and/or srHCC, relationship of HBV, HCV and C. sinensis with HCC. 

3. It would be good if the author can add more discuss about C. sinensis in HCC when compare with srHCC.

**Editorial and Data Presentation Modifications?**

Reviewer #1: No more comment.

Reviewer #2: (No Response)

Reviewer #3: (No Response)

**Summary and General Comments**

Reviewer #1: In this study, authors evaluated the association of Clonorchis sinensis infection with the risk of ruptured HCC in Guangxi, an endemic province of Clonorchis sinensis infection in China. Previously, this group characterized the association of Clonorchis sinensis infection with the risk of HCC. This is a new study not being reported. The study design is right. The outcomes are important in this field. However, minor revision is needed focusing on following aspects.

1. The paragraph "Study Population" should be re-written like this: Of 328 patients with ruptured HCC, 171 were excluded (17 with previous treatment, 95 receiving TAE or TACE, 33 without treatment, and 26 without complete follow up data). In addition, the prognosis-related clinical factors including MVI, BCLC stage, tumor size, and HBV DNA should be compared between 157 included patients and 171 excluded patients. This will let readers know if the 157 patients represent the 328 patients in total. 

2. The significance was found in patients adjusted by inverse probability of treatment weighting (IPTW). Pls discuss how this kind of matching might avoid potential bias? 

3. HBV infection between the two groups should be included in the survival analysis because HBV replication has been proven to be an independent risk factor of HCC prognosis.

Reviewer #2: Overview and general recommendation:

This study analyzed the role of C. sinensis infection in the survival outcomes of spontaneous rupture hepatocellular varcinoma patients using inverse probability of treatment weighting and found that C. sinensis infection had an adverse impact on OS in srHCC patients and was independent prognostic factor. This study was novel. However, there still some questions.

Questions and suggestions:

1. As we know, the destiny of tumors and survival outcomes of patients were decided by many factors including the pathological type, size, metastasis of tumors, chemotherapy, patient's physical condition and diet, and so on. However, these factors were not evaluated in this study. Usually, for patients with tumor of late stage, chemotherapy is needed and is a very important factor that will influence the outcomes of patients. Obviously, these factors were not designed and considered.

2. As the author said, most patients in this study were with chronic hepatitis B infection which was another very important virus that destroys the liver function. This should also be evaluated. Line 259, HCV should be HBV.

3. For OS, before IPTW, the median for the C. sinensis group was 17 months while that for the no C. sinensis group was 32 months. There was already a big difference between these two groups. However, the P was below 0.05. Why?

4. From 2013 to 2021, this study lasted for 8 years. When patients had partial hepatectomy and found to be infected with C. sinensis, did not they receive any treatment for this parasite? Any treatment will also have an impact on the results of this study.

5. C. sinensis should be italic.

Reviewer #3: The manuscript entitled “Clonorchis sinensis on the prognosis of patients with spontaneous rupture of Hepatocellular Carcinoma: an inverse probability of treatment weighting analysis” is interesting work, but there are major points need to be addressed before publish as followed; 

1. It would be better if the authors can provide the information of IPTW in the introduction part, including the advantages, and how it important and improve the result or previous reports. 

2. This finding did not show the significant data of C. sinensis and no C. sinensis group when no using IPTW, could you please explain and discussion in this point. 

3. Please Italics for the word “C. sinensis” in the text. 

4. To obvious, please describe separately between the baseline characteristics and the result that comparison of before and after using IPTW. Importantly, please use Chi-square to compare the different between before and after using IPTW. Could you please explain more about the statistical analysis of this part?

5. Please elucidate how srHCC differ from HCC, what is the significant point to study? 

6. Please add more risk factors of HCC and/or srHCC, relationship of HBV, HCV and C. sinensis with HCC. 

7. It would be good if the author can add more discuss about C. sinensis in HCC when compare with srHCC.

PLOS authors have the option to publish the peer review history of their article (what does this mean?). If published, this will include your full peer review and any attached files.

Reviewer #1: Yes: Guangwen Cao

Reviewer #2: No

Reviewer #3: No
---

## [Decision Letter · Decision Letter 1]

4 Jan 2024

Dear Dr. Xiang,

Thank you very much for submitting your manuscript "Clonorchis sinensis on the prognosis of patients with spontaneous rupture of Hepatocellular Carcinoma: an inverse probability of treatment weighting analysis" for consideration at PLOS Neglected Tropical Diseases. As with all papers reviewed by the journal, your manuscript was reviewed by members of the editorial board and by several independent reviewers. The reviewers appreciated the attention to an important topic. Based on the reviews, we are likely to accept this manuscript for publication, providing that you modify the manuscript according to the review recommendations. 

Sincerely,

Javier Sotillo

Academic Editor

Uriel Koziol

Section Editor

Reviewer's Responses to Questions

**Key Review Criteria Required for Acceptance?**

**Methods**

-Are the objectives of the study clearly articulated with a clear testable hypothesis stated?

-Is the study design appropriate to address the stated objectives?

-Is the population clearly described and appropriate for the hypothesis being tested?

-Is the sample size sufficient to ensure adequate power to address the hypothesis being tested?

-Were correct statistical analysis used to support conclusions?

-Are there concerns about ethical or regulatory requirements being met?

Reviewer #1: The methods are clearly described although their English needs to be polished.

Reviewer #2: (No Response)

Reviewer #3: Please the approval ethics number(s)/ID(s) of approval ethics.

**Results**

-Does the analysis presented match the analysis plan?

-Are the results clearly and completely presented?

-Are the figures (Tables, Images) of sufficient quality for clarity?

Reviewer #1: The results are well presented.

Reviewer #2: (No Response)

Reviewer #3: (No Response)

**Conclusions**

-Are the conclusions supported by the data presented?

-Are the limitations of analysis clearly described?

-Do the authors discuss how these data can be helpful to advance our understanding of the topic under study?

-Is public health relevance addressed?

Reviewer #1: The outcomes of this study should be reference for understanding the mechanism by which HCC recur.

Reviewer #2: (No Response)

Reviewer #3: (No Response)

**Editorial and Data Presentation Modifications?**

Reviewer #1: accept

Reviewer #2: (No Response)

Reviewer #3: (No Response)

**Summary and General Comments**

Reviewer #1: No more suggestions

Reviewer #2: The author revised the manuscript. However, there are still some should be addressed before being published. 

1. Table 1, the baseline characteristics of the study population were not complete. Such as, for age, why patients only with the age ≤50 years) were analyzed? And, like tumor size, what about patients with tumor size >10 cm? What does the size mean, length or width? All the characteristics in this table were only including some part of patient, not whole population in this study. Will not those characteristics missing in this table influence the results?

2. All C. sinensis were not in italic.

Reviewer #3: (No Response)

PLOS authors have the option to publish the peer review history of their article (what does this mean?). If published, this will include your full peer review and any attached files.

Reviewer #1: Yes: Guangwen Cao

Reviewer #2: No

Reviewer #3: No

Figure Files:

Data Requirements:

Reproducibility:

References

---

## [Decision Letter · Decision Letter 2]

12 Feb 2024

Dear Dr. Xiang,

We are pleased to inform you that your manuscript 'Clonorchis sinensis on the prognosis of patients with spontaneous rupture of Hepatocellular Carcinoma: an inverse probability of treatment weighting analysis' has been provisionally accepted for publication in PLOS Neglected Tropical Diseases.

Best regards,

Javier Sotillo

Academic Editor

Uriel Koziol

Section Editor

Reviewer's Responses to Questions

**Key Review Criteria Required for Acceptance?**

**Methods**

-Are the objectives of the study clearly articulated with a clear testable hypothesis stated?

-Is the study design appropriate to address the stated objectives?

-Is the population clearly described and appropriate for the hypothesis being tested?

-Is the sample size sufficient to ensure adequate power to address the hypothesis being tested?

-Were correct statistical analysis used to support conclusions?

-Are there concerns about ethical or regulatory requirements being met?

Reviewer #2: (No Response)

Reviewer #3: (No Response)

**Results**

-Does the analysis presented match the analysis plan?

-Are the results clearly and completely presented?

-Are the figures (Tables, Images) of sufficient quality for clarity?

Reviewer #2: (No Response)

Reviewer #3: (No Response)

**Conclusions**

-Are the conclusions supported by the data presented?

-Are the limitations of analysis clearly described?

-Do the authors discuss how these data can be helpful to advance our understanding of the topic under study?

-Is public health relevance addressed?

Reviewer #2: (No Response)

Reviewer #3: (No Response)

**Editorial and Data Presentation Modifications?**

Reviewer #2: (No Response)

Reviewer #3: (No Response)

**Summary and General Comments**

Reviewer #2: (No Response)

Reviewer #3: (No Response)

PLOS authors have the option to publish the peer review history of their article (what does this mean?). If published, this will include your full peer review and any attached files.

Reviewer #2: No

Reviewer #3: No

---

## [Editor Report · Acceptance letter]

16 Feb 2024

Dear Dr. Xiang,

We are delighted to inform you that your manuscript, "Clonorchis sinensis on the prognosis of patients with spontaneous rupture of Hepatocellular Carcinoma: an inverse probability of treatment weighting analysis," has been formally accepted for publication in PLOS Neglected Tropical Diseases.

Best regards,

Shaden Kamhawi

co-Editor-in-Chief

Paul Brindley

co-Editor-in-Chief
